# Degradation of P(3HB-*co*-4HB) Films in Simulated Body Fluids

**DOI:** 10.3390/polym14101990

**Published:** 2022-05-13

**Authors:** Juraj Vodicka, Monika Wikarska, Monika Trudicova, Zuzana Juglova, Aneta Pospisilova, Michal Kalina, Eva Slaninova, Stanislav Obruca, Petr Sedlacek

**Affiliations:** Faculty of Chemistry, Brno University of Technology, Purkynova 118, 612 00 Brno, Czech Republic; xcvodickaj@vutbr.cz (J.V.); monika.wikarska@vut.cz (M.W.); xctrudicova@fch.vut.cz (M.T.); zuzana.juglova@vut.cz (Z.J.); xcpospisilovaan@fch.vut.cz (A.P.); kalina-m@fch.vut.cz (M.K.); xcslaninovae@fch.vut.cz (E.S.); obruca@fch.vut.cz (S.O.)

**Keywords:** polyhydroxyalkanoates (PHA), *Aneurinibacillus* sp. H1, PHA copolymers, biodegradation, simulated body fluids

## Abstract

A novel model of biodegradable PHA copolymer films preparation was applied to evaluate the biodegradability of various PHA copolymers and to discuss its biomedical applicability. In this study, we illustrate the potential biomaterial degradation rate affectability by manipulation of monomer composition via controlling the biosynthetic strategies. Within the experimental investigation, we have prepared two different copolymers of 3-hydroxybutyrate and 4-hydroxybutyrate—P(3HB-*co*-36 mol.% 4HB) and P(3HB-*co*-66 mol.% 4HB), by cultivating the thermophilic bacterial strain *Aneurinibacillus* sp. H1 and further investigated its degradability in simulated body fluids (SBFs). Both copolymers revealed faster weight reduction in synthetic gastric juice (SGJ) and artificial colonic fluid (ACF) than simple homopolymer P3HB. In addition, degradation mechanisms differed across tested polymers, according to SEM micrographs. While incubated in SGJ, samples were fragmented due to fast hydrolysis sourcing from substantially low pH, which suggest abiotic degradation as the major degradation mechanism. On the contrary, ACF incubation indicated obvious enzymatic hydrolysis. Further, no cytotoxicity of the waste fluids was observed on CaCO-2 cell line. Based on these results in combination with high production flexibility, we suggest P(3HB-*co*-4HB) copolymers produced by *Aneurinibacillus* sp. H1 as being very auspicious polymers for intestinal in vivo treatments.

## 1. Introduction

With the reputation of being green polymers, polyhydroxyalkanoates (PHA) represent a very “hot” alternative to commercial polymers. PHA represents an extensive family of biopolymers, including homopolymers and copolymers, which are very diverse in monomer composition [1]. Unlike petrochemical polymers, PHA are entirely biodegradable and biocompatible. Their physico-mechanical properties evolve from the monomer composition and often balance the properties of conventional plastics. Amongst the PHA group, copolymers containing 4-hydroxybutyrate are recently being increasingly investigated due to their suitability for various medical applications [2].

A wide range of bacterial strains are able to decompose PHA, simultaneously producing several types of highly specific PHA-depolymerases [3,4]. However, enzymatic degradation of polyhydroxyalkanoates could also be ensured by the action of other esterases, e.g., lipases [5]. Due to the ubiquity of lipases, a wide spectrum of organisms are expected to be able to decompose PHA. According to Mok et al., lipases from pancreatic extracts from mice and chicken indicated similar PHA-depolymerizing activity on copolymer P(3HB-*co*-4HB) compared to several microbial lipases [6]. The decomposition of poly(4-hydroxybutyrate) films by bacterial (*Pseudomonas cepacia*) and fungal (*Rhizopus oryzae*) lipase in simulated physiological conditions was examined by Keridou et al., and their data demonstrated notable differences between enzymatic and abiotic hydrolysis [7,8]. In addition, the authors indicated different mechanisms of hydrolysis in the presence of lipases compared to their absence.

The rate of PHA-degradability depends on the crystal morphology—whilst highly crystalline regions are more resistant to hydrolysis, amorphous parts are not so stable and might undergo very quick hydrolysis [3,9,10]. Since the homopolymer of 3-hydroxybutyrate (3HB) occurs in high-crystallinity form, various PHA copolymers appear more amorphous and thus much more prone to hydrolytic degradation. Moreover, higher amorphousness stands for enhancement of some of the material properties (higher elongation at break and lower melting and glass transition temperature), therefore ensuring higher flexibility of use [11,12]. One of the most promising copolymers with promoted mechanical properties is P(3HB-*co*-4HB). Combining in vivo biodegradability and biocompatibility, this copolymer is the auspicious candidate for in vivo applications in medicine [13]. There are only a few described producers able to incorporate 4-hydroxybutyrate into the polymer chain. As Pernicova et al. reported previously, *Aneurinibacillus* sp. H1 is a unique bacterial producer of P(3HB-*co*-4HB) able to produce this copolymer containing various 4-hydroxybutyrate (4HB) portions according to the selected initial 1,4 butanediol:glycerol ratio [14]. Therefore, the properties of the final product are possibly adjustable by setting the cultivation conditions.

Besides the material properties, the key characteristics with respect to medical uses of biopolymers such as PHA also lie in its degradability/stability under physiological conditions. We consider this crucial behavior of these frequently investigated materials as insufficiently described in the term of biomedical applications. The future prospects set up requirements for the degradation course exploration in the body-simulating environment, following a major degradation rate and mechanisms. Therefore, in this study, we prepared films of two various copolymers of P(3HB-*co*-4HB) differing in monomer ratios and compared their biodegradation in synthetic human gastric juice (SGJ) and artificial colonic fluid (ACF) to degradation of reference homopolymer (P3HB) in order to assess the benefits of application of PHA copolymers in biomedical technology. We have also performed unique attempts to evaluate the presence of degradation products and its potential health risks via cytotoxicity tests.

## 2. Materials and Methods

### 2.1. Preparation of Polymer Films

Production of P(3HB-*co*-4HB) copolymers was performed by the cultivation of thermophilic bacterium *Aneurinibacillus* sp. H1, as reported previously [14]. The inoculum was generated by incubating the cells in complex medium Nutrient Broth (10 g/L beef extract, 10 g/L peptone, 5 g/L NaCl; HiMedia, Maharashtra, India) at 45 °C, 180 rpm, for 24 h. The cells were subsequently inoculated to mineral salt medium (MSM) in the ratio of 10%. MSM consisted of Na_2_HPO_4_·12 H_2_O, 9.0 g/L; KH_2_PO_4_, 1.5 g/L; MgSO_4_·7 H_2_O, 0.2 g/L; NH_4_NO_3_, 1.0 g/L; CaCl_2_·2 H_2_O, 0.02 g/L; FE^III^NH_4_citrate, 0.0012 g/L; Tryptone, 0.5 g/L and microelement solution (MES, composition see Appendix A), 1 mL/L (all chemicals from Lach-ner, Neratovice, Czech Republic). Two ratios of carbon sources were used: 4:6 g/L of 1,4-BD: glycerol (lower 4HB content), 4:2 g/L of 1,4-BD: glycerol (higher content of 4HB in the copolymer). The production medium was incubated in shaking flasks (45 °C, 180 rpm) for 72 h. In order to produce a homopolymer of 3HB, *Cupriavidus necator* was employed. The inoculum was generated analogically to *Aneurinibacillus*, however, the incubation temperature was 30 °C. The MSM consisted of KH_2_PO_4_, 1.02 g/L; (NH_4_)_2_SO_4_, 1.0 g/L; Na_2_HPO_4_·12 H_2_O, 11.1 g/L, MgSO_4_·7 H_2_O, 0.2 g/L; trace element solution (TES, composition see Appendix A), 1 mL/L; as the source of carbon, fructose (20 g/L) was used (all from Lach-ner, Neratovice, Czech Republic). After the incubation, the bacterial cultures were centrifuged at 4800× *g* for 10 min, and the obtained biomass was dried at 70 °C in Petri dishes. The monomer composition and overall PHA content were determined by Gas Chromatography method according to Obruca et al. [15]. As an isolation step, chloroform extractions of the polymers from biomass were held by using SOXTHERM^®^ automatic extraction system (C. Gerhardt Analytical systems, Königswinter, Germany).

To generate polymer films, materials obtained were redissolved in chloroform (5% *w*/*w*) and solutions were filtered through 5 µm-nylon filters. An automatic film applicator (TQC Sheen, Capelle aan der IJssel, The Netherlands) was used in order to prepare the films, where each film consisted of two layers applied in opposite directions. To determine film thickness, a mechanical profilometer (Bruker, Billerica, MA, USA) was employed. All chemicals used in this study reached A.R. grade.

### 2.2. Biodegradation in Simulated Body Fluids

Biodegradation of manufactured films was held in two simulated body fluids (SBFs)—simulated gastric juice (SGJ—pH 1.6) and artificial colonic fluid (ACF—pH 7.8); and phosphate buffer solution (PBS—0.01 M, pH 7.4) to monitor abiotic hydrolysis in a water environment. The SBFs were prepared according to Marques et al. (see ref. [16]); the compositions are available in Appendix A. The films were cut into 2.5 × 2.5cm specimens, which were rinsed in ethanol to prevent medium contamination and then dried at room temperature. Specimens were incubated for 12 weeks at 37 °C and 160 rpm at an orbital incubator shaker. Fluids were refreshed weekly; samples of recycled fluids were held in the freezer for further analyses. After 2, 4, 8, and 12 weeks of incubation, triplicates of films were rinsed in ethanol, dried at room temperature, and weighed for the following analyses.

### 2.3. Molecular Weight Measurements

Molecular weights of polymers were analyzed as previously [17]. In short, 1.5 mg of individual polymer samples were dissolved in 1 mL of chloroform (HPLC-grade) and subsequently passed through nylon membrane filters (0.45 µm). A total of 100 µL of each sample was injected to the size exclusion chromatography system (Infinity 1260 system with PLgel Mixed-C column, Agilent, Santa Clara, USA), for the detection, multi-angle light scattering (Dawn Heleos II, Wyatt Technology, Santa Barbara, CA, USA) and differential refractive index (Optilab T-rEX, Wyatt Technology, Santa Barbara, CA, USA) detectors were employed. As a mobile phase, HPLC-grade chloroform (filtered through 0.02 µm membrane filters) at a flow rate 0.6 mL/min was used. To determine the weight-average molecular weight (M_w_) and polydispersity index (PDI), ASTRA software (version 7.3.2, Wyatt Technology, Santa Barbara, CA, USA) was used employing Zimm equation.

### 2.4. DSC Measurements

DSC measurements were provided according to previous method [17]. Briefly, change in crystallinity after 12-weeks incubation in SBFs was investigated by DSC measurements using a calorimeter (DSC 2500, TA Instruments, New Castle, DE, USA). Samples were analyzed in hermetically sealed TzeroTM aluminum pans (TA Instruments, New Castle, DE, USA) under a dynamic nitrogen atmosphere. Single-heating program (ramp 10 °C/min to 195 °C) and cooling (ramp 10 °C/min to 30 °C) was applied. As a calibrant, indium was used.

### 2.5. Detection of Degradation Products

Samples of fluids were investigated for the residual 3-hydroxybutyric acid and crotonic acid content as the putative products of polymer decomposition. The ion-exchange chromatography (IC) system (Metrohm, Herisau, Switzerland) coupled with Metrosep Organic Acids—250/7.8 column (Metrohm, Herisau, Switzerland) and conductivity detector (850 Professional IC Anion, Metrohm, Herisau, Switzerland) were employed for this purpose. For the isocratic elution, 0.5 mM HClO_4_ with 10 mM LiCl was used. The samples of SGJ were pre-neutralized by 0.01 M NaOH. All samples were pre-filtered through 0.45 µm syringe filters (nylon membrane).

### 2.6. Surface Imaging

The small specimens from each sample were gold-coated in a sputtering coater (Polaron) and investigated using a ZEISS EVO LS 10 scanning electron microscope (Carl Zeiss NTS, Munich, Germany). Observations were realized in secondary electron (SE) mode, andthe accelerating voltage was 5 kV (avoiding sample charging). The method corresponds to a previously used SEM procedure by Sedlacek et al. [17].

### 2.7. Cytotoxicity Tests

Cell viability assay study for determination of cytotoxicity was performed using Human Colon Adenocarcinoma cell line, permanent cell line CaCO-2 (Cell Lines Service, Eppelheim, Germany, SRN). The cells were grown in Eagle’s Minimum Essential Medium—MEM supplemented with 10% fetal bovine serum, 1% Antibiotic-Antimycotic blend containing 100 mg/mL of streptomycin, 100 units/mL of penicillin and 0.25 µg/mL of Gibco Amphotericin B in humidified, 5% (*V/V*) CO_2_ atmosphere at 37 °C. Every second day, the medium was refreshed, and the cells were eventually subcultured at 80–90% confluency by treatment with 0.25% trypsin-EDTA. For the MTT assay, the cells were seeded at a density of 1800 cells per well (100 µL) and cultivated overnight. The samples obtained by polymers foil digestion from weeks 2, 4, 8, and 12 were filtered through a 200 nm PET syringe filter. The growth medium was replaced after 24 h of seeding and the samples were added in the concentration range 25%, 12%, 6.5%, and 3.25%, using ethanol as a negative control and growth media as a control culture. After 24 h of cultivation, 20 µL of MTT dissolved in PBS (2.5 mg/mL) was added to each well and the plate was placed for 3 h in cultivator (BioTek, Winooski, VT, USA). Then, 100 µL of 10% SDS in PBS was added and the plate was stored in laboratory temperature until the next day. The evaluation was performed the next day by ELISA Reader (Synergy HTX Multimode reader, BioTek, Winooski, VT, USA) at 562 nm. The test design was inspired by Li et al. [18].

### 2.8. Statistical Analysis

All results are presented as mean values from triplicates. Statistical comparisons were provided by one-way analysis of variance (ANOVA), and differences were considered significant at *p* < 0.05. All statistical analyses were provided in Statistica^®^ 13.3 software (TIBCO, Palo Alto, CA, USA).

## 3. Results and Discussion

### 3.1. Weight Changes

Parameters of manufactured films are presented in Table 1. The specimens for degradation test and its macroscopic morphology at the end of incubation are shown in Figure 1. Within the first four weeks of the test, no significant mass changes were observed, excluding P34HB-2 in colonic fluid (Figure 2). On the contrary, molecular weight decreases are noticeable at this point, especially in the case of gastric juice (Table 2). In the later stage of the test, films of both copolymers became more brittle and appeared fragmented. A substantial part of the material disappeared. Moreover, some smaller fragments were lost due to the manipulation difficulties during the fluid refreshment, which most likely stands for markedly higher deviations. Additionally, only large-scale cracks appear on the SEM micrographs (see Figure 3), and no surface erosion is observed. Based on these results, it is obvious that the low pH of the gastric juice might induce abiotic hydrolysis, resulting in polymer cracking.

The weight of the samples is generally expected to decrease during degradation. Nevertheless, it was reported that the change in weight is not a reliable indicator for the degree of PHA degradation evaluation. Based on available studies, degradation of PHA specimens shows lag-phase in weight loss or very mild development of the mass decrease in different environments [19,20,21,22,23]. The process is possibly initiated by the scission of macromolecules to shorter chains and oligomers, while the mass of a specimen undergoes only slight fluctuations. The shorter polymer chains are still too hydrophobic to dissolve in an aqueous environment, so they remain adhered to the matrix until the length of oligomers is reduced enough for the mass release into environment. For this reason, changes in molecular weight are much more evident than weight changes. This phenomenon has already been reported by several authors. Han et al. investigated biodegradation of copolymer P(3HB-*co*-3HV) films in a phosphate buffer solution with porcine pancreas lipase; after 180 days of incubation, the authors observed only slight weight deflections (remaining more than 95% of initial weight), while the molecular weight decrease reached 34–80% [24]. Zhuikov et al. reported similar results as various PHA copolymers incubated in the fluid containing pancreatic lipase, where the mass decreased by 8–9% and the molecular weight reduction reached up to 80% [25].

Compared to the copolymers, P3HB did not exhibit any significant weight changes during the whole incubation in any fluid (Figure 2). These results correlate to outputs presented by Kovalcik et al., who observed a weight reduction of 3.6% of PHB scaffolds in synthetic gastric juice and lipase-supplemented PBS after 81 days of incubation [26]. We report an average mass decrease of 1.5% and 2.5% in gastric juice and colonic fluid after 12 weeks of incubation, respectively. The 3HB-homopolymer is obviously more resistant to degradation processes, so it is predisposed to quite different biomedical applications compared to copolymers, where its prolonged disintegration is beneficial. Similar results were obtained by Chuah et al. who observed the biodegradation of various PHA in PBS with porcine pancreas lipase [27]. The author reported complete degradation of P(98 mol% 4HB-*co*-3HB) films, while no degradation of P3HB films was detected [27].

Additionally, our results show large deviations due to coagulation of the fluid components (proteins and bile salts) on the films, especially in the colonic fluid. Even though these components were rinsed off the specimens until visual absence, some particles persisted on the surface and inside the pores of the homopolymer.

### 3.2. Molecular Weight Changes

As stated before, changes in molecular weight offer a more reliable view on the biodegradation process than simple weight change monitoring. The course of the molecular weight changes is shown in Table 2. In general, the largest Mw losses were detected in simulated gastric juice, where all samples reached 80% Mw reduction. Since this medium did not contain any esterases, we do not suppose enzymatic hydrolysis as a reason of this Mw reduction. Pepsinolysis also seems improbable as far as pepsin is able to digest peptide bonds nearby large-side chain amino acids (no substrate analogy). Hence, acid-catalyzed hydrolysis appears as the most probable source of chain scission. Simple hydrolysis of the samples is also noticeable in PBS medium. While the Mw losses are quite mild in a comparison with films incubated in SGJ, the influence of low pH in SGJ is more observable. As SGJ is a substantially aggressive environment for polyester incubation, it represents a medium suitable for rapid decomposition of P(3HB-*co*-4HB). On the contrary, incubation in ACF showed slightly consistent molecular weight decreases, so this medium suggests a different degradation mechanism. Nevertheless, the presence of enzymes (pancreatin) obviously accelerates hydrolysis. This is also supported by Freier et al. [28].

Interestingly, P3HB reached nearly the same final degree of chain scission (approximately 20% of initial M_w_) as the copolymers in SGJ. Commonly, in ACF, the homopolymer almost reached the highest M_w_ loss amongst the tested PHAs. The very analogic courses in both SBFs of decomposition suggest overall aggressivity of SGJ and ACF environments, possibly deleting the crystal-morphological handicap of homopolymer. Similar results were also obtained by Kovalcik et al. while degrading PHB and P(3HB-*co*-3HV) in SGJ and PBS with microbial lipase [26]. The authors reported an even higher rate of M_w_ decreases in PHB than in copolymers and suggest an explanation, stating that the decrease-rate differences are disproportionally lower than M_w_ differences between polymers. This hypothesis is based on differences in the probability of hydrolysis, in which a larger set of hydrolysis targets (ester bonds) provides a higher probability of hydrolysis, which might explain the phenomena of our tests. On the other hand, PBS showed a significant difference between incubated specimens, where P3HB showed a very weak tendency to lose molecular weight (approximately 20% loss) compared to copolymers. Likewise, Doi et al. observed different rates of M_w_ decrease as incubating P(3HB-*co*-4HB) in PBS, depending on monomer composition [29]. Indeed, simple hydrolysis was in progress, however, the environment did not offer more efficient catalytic sources, so it was insufficient for the decomposition of highly crystalline regions of the homopolymer. The authors further reported that random hydrolytic scission acted through the whole polymer matrix [29].

Surprisingly, an increase of P3HB molecular weight was observed in PBS after 2 weeks of incubation. The exact reason of this phenomenon is not obvious within the scope of this work. Two possible explanations are offered: the leaching of the short polymer chains, which is inconsistent with the mass decrease results; or spontaneous polycondensation, which is unfavorable in such an environment [30,31]. The leaching of low-molecular polymer fractions was previously discussed also by Zhuikov et al. [25].

### 3.3. Changes in Thermal Properties

DSC thermograms of the initial state of polymers are available in the Appendix A. Data from DSC showed several fluctuations in melting enthalpy after incubation in SBFs. Since the increase of ΔH_m_ is associated with the rise of the degree of crystallinity, we observed two significant increases (alpha significance level of 0.05) of melting enthalpy (see Table 3) P3HB in SGJ, and 4HB component of P34HB–2 in SGJ. Further, changes in ΔH_m_ are apparent also in 4HB crystals of P34HB–1 after the incubation in SGJ, however, the statistical analysis was not provided for the data insufficiency (the asterisks in Table 3). Commonly, the stated shifts probably result from the physico-chemical characteristics of the environment, mostly from the low pH of the SGJ. As is generally known, highly crystalline regions provide higher hydrolysis resistance as compared to amorphous phases [32]. Accordingly, the amorphous phase might have preferably been subjected to degradation processes, which thus led to an overall increase of crystallinity. The same trend was observed by Keridou et al. while degrading P4HB films [8]. Also, this result is consistent with the degradation of other biopolyesters, for instance PLA [33].

### 3.4. Detection of Degradation Products

According to the weight-changes results, we have expected negligible amounts of 3-hydroxybutyric and crotonic acid. Therefore, only samples reporting the highest weight losses were chosen for the IC analysis. Unfortunately, not even the method of standard addition revealed any content of analytes in the studied matrices. The amounts of released acids most likely lie under the limits of detection. A typical chromatogram is illustrated in Appendix A. These results confirm the previous hypothesis of the degradation mechanism of PHA (discussed in the weight changes section), indicating that polymer specimens do not show any significant weight losses within the initial phase of degradation, whereas the molecular weight changes are apparent.

### 3.5. Surface Changes

As shown in Figure 3, the surface of poly(3-hydroxybutyrate) did not undergo any significant changes during the course of incubation. The pore diameters and frequency were not increased in any of the fluids after the performance of the full test. The initial crystallinity of the homopolymer (corresponding to ΔH_m_ 75.4 J/g, see Table 3) apparently slowed down the hydrolytic activity of the environment, so the less numerous amorphous regions stayed included in the prevailing crystalline regions.

On the contrary, both copolymers showed notable changes in surface morphology, especially while incubated in ACF, as seen in Figure 3. From the fourth week of incubation in ACF, small cracks filled with submicron globules appeared (red arrows), and their extent raised in the course of degradation. At the end of incubation, about half of the surface of P34HB–1 was covered by these bodies. These globules are potentially uncovered highly crystalline regions that resisted degradation, while the mostly amorphous areas were released. In addition, chains of globules resembling pathways were attached to the surface of P34HB–1 after 4 weeks in ACF (see yellow arrows). This might be the key to explaining the specimen degradation mechanism. Since the surface appears very consistent and largely smooth, the typical activity of hydrolytic enzymes causes surface erosion, resulting in massive irregular porosity [34]. A similar scenario is observable in the later stages of degradation. Hence, enzymolysis cannot be excluded, however, it is quite clear that enzymes are not the only source of the degradation course. As mechanical contributors, shear forces may act as an initiator of material disintegration. By attacking the specimen surface, the film could raise hydrolysis susceptibility, which facilitates the spatial enzymolysis and enhances the exposure of crystalline particles. Concurrently, we presuppose that film contains a highly crystalline surface layer standing for both high rigidity and hydrolytic stability. Subsequently to the sub-surface layer degradation, the surface crystalline coating was peeled off the specimen. Due to its hydrolytic resistance, these evolved surface fragments predominantly consist of polymeric and oligomeric chains, while barely undergoing any biodegradation processes. Therefore, the absence of detectable 3-hydroxybutyrate in waste fluids and simultaneous mass reduction of the specimens is justified. Compared to P34HB–2, P34HB–1 is conclusively more prone to enzymatic hydrolysis (surface erosion).

When related to other materials, the degradation of P(3HB-*co*-4HB) copolymers in simulated gastral systems is apparent on the surface and macroscopic morphology. For example, the surface of PCL seems less affected by enzymatic hydrolysis than the presented copolymers here, even if the mass decreases are significant in both cases [35]. The surface erosion of PCL-based copolymers in the presence of pancreatic lipase was concluded by Peng et al. [36].

### 3.6. Cytotoxicity Tests

The cytotoxicity testing results are presented (Figure 4) on 25% concentration of the ACF (containing lipases) and PBS to monitor the cytotoxicity of the potential degradation products of studied polymers. As expected, the samples from all tested weeks showed no significant difference (at 95% confidence level) related to pure ACF/PBS, excluding P34HB-1 in ACF sampled on the 8th week, which increased the cell viability from 53% (pure ACF) to 86%. Accordingly, neither PBS nor ECF after incubation indicate any inhibition effects on the cell cultures. Moreover, potential copolymer residues might beneficially support the cell growth. These outputs correlate to the theory of the degradation mechanisms discussed above, since the increase of the cell viability appeared as late as the significant mass decreases. Presumably, according to these results, the P34HB–1 film released higher amounts of the water-soluble degradation products during the 8th week of incubation (monomers, short oligomers), which are possibly utilized by the cell line. To summarize, the degradation products of the tested polymers did not reveal cytotoxic effect with respect to used cell line and rather supported growth of the cell culture, which stresses the suitability of the tested materials for biomedical applications.

## 4. Conclusions

In this study, we have tested the biodegradability/stability of PHB and P(3HB-*co*-4HB) copolymers in various simulated body fluids. All the tested materials exhibited evident degradability in simulated body fluids. According to the presumption, copolymers showed markedly higher promptness to biodegradation compared to a homopolymer, as observed on surface morphology. On the other hand, results of further analyses also demonstrated the degradation processes in P3HB films (molecular weight changes, changes in melting enthalpy). No toxic products of the material degradation in waste SBFs were observed on cell viability. Additionally, 3-hydroxybutyrate as a free monomer was not detected by ion-exchange chromatography. It is likely that degradation products consist predominantly of water-soluble oligomers, which were not detected by the chromatographic analysis. Within the fluids used for incubation, synthetic gastric juice appeared as the most aggressive environment due to its low pH, which resulted in the highest weight reductions of both copolymers. Most likely, this weight loss cannot be attributed to enzymatic hydrolysis, and we suggest solely an abiotic consequence of degradation. Artificial colonic fluid appears as a very promising environment for the P(3HB-*co*-4HB) copolymers biodegradation, apparently causing surface erosion of this material. Possible mechanisms of decomposition are subject to further investigation.

Copolymers of 3-hydroxybutyrate and 4-hydroxybutyrate are evidently very auspicious materials for in vivo applications, especially for the treatments requiring short-term biodegradability compared to simple 3-hydroxybutyrate homopolymer, which pose as more stable against degradation. Copolymers appear very promising for intestinal therapy, where P(3HB-*co*-4HB) copolymers are biodegradable and thus possibly prevent the release of undesirable cytotoxic products. Accordingly, further in-depth research for released oligomers detection is needed. On the contrary, P3HB shows higher degradation stability and fits biocompatible materials, ensuring low rate intracorporeal degradability. The rate of biodegradation and simultaneous biocompatibility are key factors to select for the in vivo intestinal applications [37,38]. As far as we are able to manipulate with the copolymer degradation rate, these materials appear as very promising substituents of other investigated materials such as PLA or PCL [33,35]. Obviously, the results of this study confirm that, in the family of PHA polymers, the wide range of potential biomedical application with specific requirements on the rate of degradation can be covered simply by adjustment of the appropriate monomer composition.

There are still many economic challenges and limitations of the PHA biotechnological production, which are mainly affected by relatively high production costs and the difficulty of the extraction steps when compared to other biopolymers. Nevertheless, many research groups currently investigate novel production approaches and cultivation strategies in order to increase the competitiveness of these highly promising biopolymers [39,40,41,42]. The study presented here is consistent with this concept as far as the thermophilic (bacterial) PHA producent has been employed, which suggests the potential reduction of sterility requirements. The introduced aspects should improve the competitiveness of PHA copolymers and extend the applicability spectrum. Furthermore, in-depth exploration of structural and functional properties of the presented materials is suggested for further production and development.

## Figures and Tables

**Figure 1 polymers-14-01990-f001:**
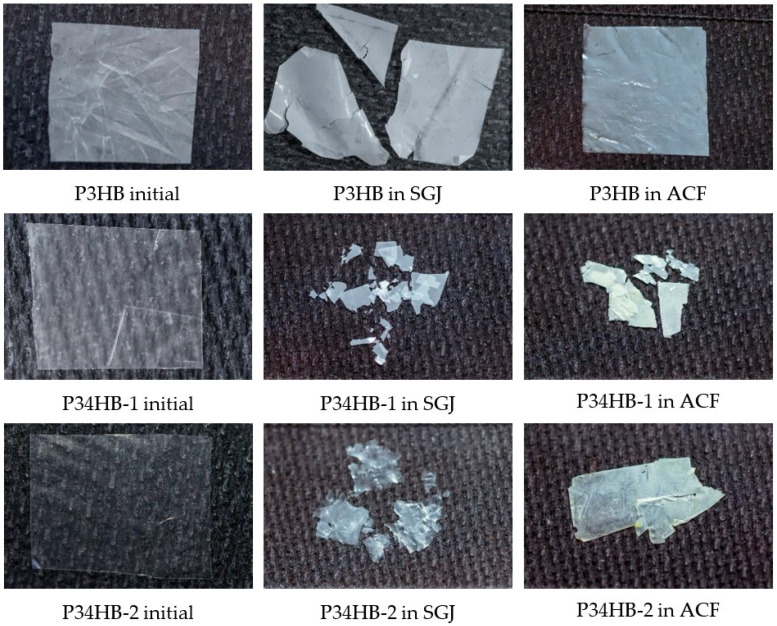
Changes in the morphology of the films at the end of incubation in SBFs.

**Figure 2 polymers-14-01990-f002:**
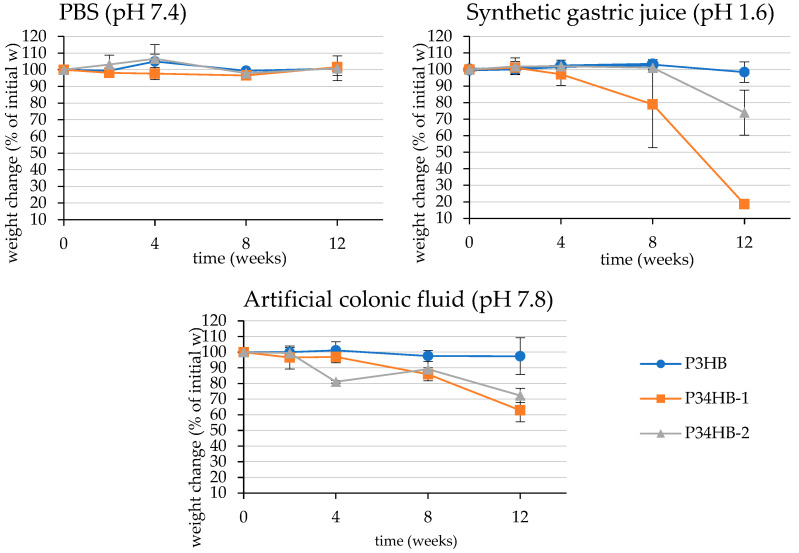
Weight changes of films during incubation in PBS, synthetic gastric juice, and artificial colonic fluid, expressed as relative weight change in %.

**Figure 3 polymers-14-01990-f003:**
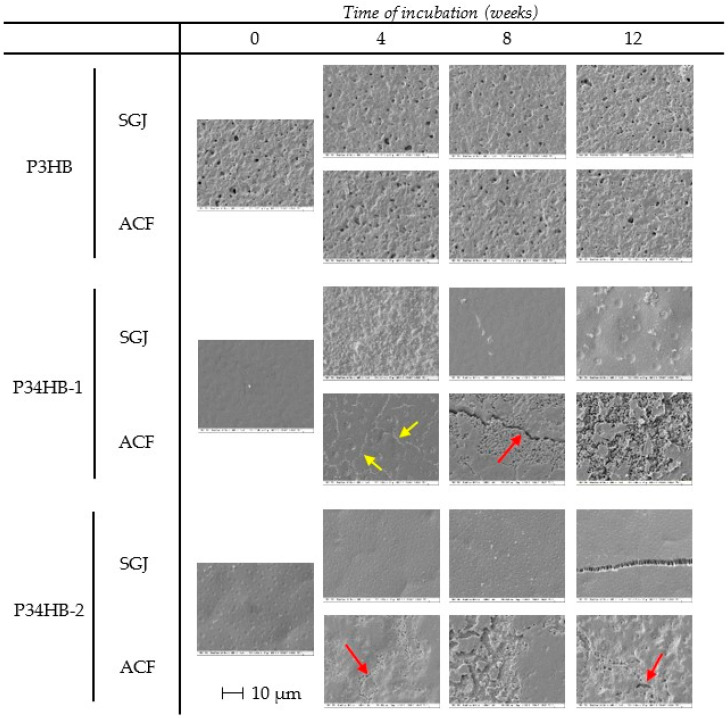
SEM micrographs showing surface morphology changes during incubation in SBFs.

**Figure 4 polymers-14-01990-f004:**
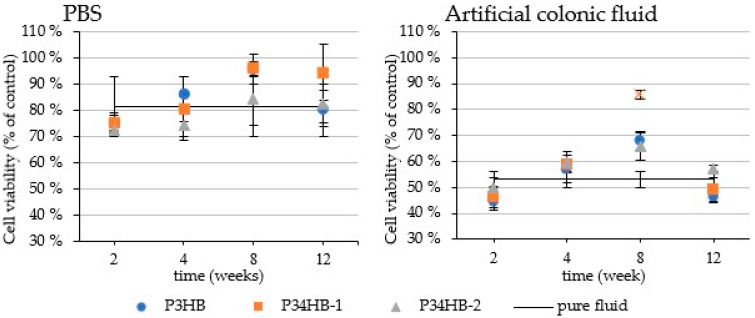
The influence of PBS and ACF after polymers incubation on the growth of Caco-2 cell culture expressed as percentual viability related to the fresh fluids. The data values (presented as means ± SD) are expressed as percentual viability related to the control (pure medium). A statistically significant difference (*p* < 0.05) is indicated by a cross (✕).

**Table 1 polymers-14-01990-t001:** Parameters of manufactured films.

Polymer	4HB (mol.%)	Film Thickness (µm)	Mw (kDa)	Mn (kDa)	PDI (–)
P3HB	0	11.0 ± 1.1	481.26 ± 11.62	380.1 ± 23.47	1.27 ± 0.07
P34HB–1	36	11.1 ± 1.4	127.14 ± 1.73	72.83 ± 7.18	1.76 ± 0.20
P34HB–2	66	11.1 ± 0.6	174.13 ± 4.27	134.0 ± 5.38	1.3 ± 0.03

**Table 2 polymers-14-01990-t002:** Changes in molecular weight expressed as percentual weight of initial M_W_.

Fluid	Polymer	Initial	2. Week		4. Week		8. Week		12. Week	
		M_W_ (kDa)	M_W_ (kDa)	% of Initial M_W_	M_W_ (kDa)	% of Initial M_W_	M_W_ (kDa)	% of Initial M_W_	M_W_ (kDa)	% of Initial M_W_
PBS	P3HB	481.26 ± 11.62	537.91 ± 3.59	111.77	411.07 ± 22.16	85.42	395.04 ± 18.26	82.08	385.10 ± 10.40	80.02
	P34HB-1	127.14 ± 1.73	107.29 ± 7.30	84.39	112.38 ± 4.46	88.39	96.85 ± 3.93	76.18	63.10 ± 1.00	49.63
	P34HB-2	174.13 ± 4.27	190.40 ± 3.03	109.34	164.62 ± 12.42	94.54	153.76 ± 2.07	88.30	110.60 ± 1.30	63.52
SGJ	P3HB	481.26 ± 11.62	318.81 ± 3.12	66.24	338.97 ± 13.28	70.43	176.41 ± 7.09	36.66	107.00 ± 5.30	22.23
	P34HB-1	127.14 ± 1.73	88.66 ± 3.76	69.73	57.33 ± 1.65	45.09	50.77 ± 4.46	39.93	26.30 ± 1.30	20.69
	P34HB-2	174.13 ± 4.27	110.89 ± 3.05	63.68	68.38 ± 6.27	39.27	50.27 ± 3.75	28.87	32.90 ± 0.10	18.89
ACF	P3HB	481.26 ± 11.62	462.14 ± 9.02	96.03	356.40 ± 9.97	74.06	226.14 ± 17.04	46.99	164.40 ± 5.00	34.16
	P34HB-1	127.14 ± 1.73	97.84 ± 7.64	76.95	101.74 ± 5.14	80.02	79.10 ± 8.09	62.21	50.90 ± 1.50	40.03
	P34HB-2	174.13 ± 4.27	162.55 ± 11.31	93.35	131.20 ± 3.67	75.35	97.82 ± 1.88	56.18	70.10 ± 1.50	40.26

**Table 3 polymers-14-01990-t003:** Melting enthalpy of the initial and final state of the films.

Polymer		ΔHm (J/g)			
		Initial	PBS	SGJ	ACF
P3HB		75.4	83.9 ± 2.5	87.4 ± 0.8 +	78.3 ± 1.0
P34HB–1	3HB	22.7	20.1 ± 0.4	20.7 *	23.0 ± 1.4
	4HB	22.2	26.3 ± 2.2	31.5 *	22.3 ± 1.6
P34HB–2	3HB	11.0	9.3 ± 0.3	9.5 ± 0.5	12.3 ± 0.7
	4HB	32.5	39.3 ± 2.8	47.9 ± 3.06 +	36.1 ± 0.8

* insufficient amount of sample to analyze all triplicates; + significant change (*p* < 0.05) related to initial value.

## Data Availability

The data that support the findings of this study are available from the corresponding author, P.S., upon reasonable request.

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
