# Peer review of "Degradation of P(3HB-co-4HB) Films in Simulated Body Fluids"

_polymers, 2022, doi:10.3390/polym14101990_

Round 1
Reviewer 1 Report
- The materials section is missing. Please include the brand and specifications of all the materials that are used in this study.
- The methodology section needs a reference for each, such as 2-3 to 2-7.
- The caption of Figure 2 is incomplete. It must be contained with all the three performed figures and their descriptions.
- In figure 2, please mention the pH of each fluid near the related graph for better comparison.
- In the results section, all the results of this study must be compared with other studies' results.
- It is suggested to add a ”future prospects” section at the end of the manuscript.
- There are some too old references (before 2010), it can be better if the authors replace them with newly published papers.
- The authors can use the following references in this manuscript:
Muhamad, I. I., Sabbagh, F. A. R. Z. A. N. E. H., & Karim, N. A. (2017). Polyhydroxyalkanoates: A valuable secondary metabolite produced in microorganisms and plants. Plant Secondary Metabolites, Volume Three: Their Roles in Stress Eco-Physiology, 185.
Mojaveryazdia, F. S., Zainb, N. A. B. M., & Rezaniac, S. (2013). Production of biodegradable polymers (PHA) through low cost carbon sources: Green Chemistry. International Journal, 4(3).
Sabbagh, F., & Muhamad, I. I. (2017). Production of poly-hydroxyalkanoate as secondary metabolite with main focus on sustainable energy. Renewable and Sustainable Energy Reviews, 72, 95-104.
Author Response
Dear reviewer,
Thank you for Your time taken for reviewing the submitted manuscript entitled Degradation of P(3HB-co-4HB) films in Simulated Body Fluids, we appreciate the objectivity of Your comments and suggestions, and found the reviewed manuscript much improved.
We have properly considered each point of Your comments and hereby suggest accordingly modified the manuscript. Additionally, we would like to clarify (explain) some of Your points:
Point 7, 8: There are some too old references (before 2010), it can be better if the authors replace them with newly published papers.
The authors can use the following references in this manuscript:
Muhamad, I. I., Sabbagh, F. A. R. Z. A. N. E. H., & Karim, N. A. (2017). Polyhydroxyalkanoates: A valuable secondary metabolite produced in microorganisms and plants. Plant Secondary Metabolites, Volume Three: Their Roles in Stress Eco-Physiology, 185.
Mojaveryazdia, F. S., Zainb, N. A. B. M., & Rezaniac, S. (2013). Production of biodegradable polymers (PHA) through low cost carbon sources: Green Chemistry. International Journal, 4(3).
Sabbagh, F., & Muhamad, I. I. (2017). Production of poly-hydroxyalkanoate as secondary metabolite with main focus on sustainable energy. Renewable and Sustainable Energy Reviews, 72, 95-104.”
Response 7, 8: Some old references used in the manuscript are irreplaceable since represent original source of referred information, capture facts and dogma, based on which we formulate differences (advances) and outputs. Some of those old references were substituted by newly published papers as proposed, however, several papers published between 2000 and 2010 were kept for the results comparison and potential progress evaluation. Thank You for the references recommended, some of the stated citations were applied.
Resting points have been implemented directly to the manuscript. Also, the formal site and English has been checked and corrected by native speaker. Please, kindly find the modified manuscript enclosed.
We hope that provided modifications and offered explanations will meet Your requirements.
Best regards,
The authors collective

Reviewer 2 Report
I read carefully the article entitled Degradation of P(3HB-co-4HB) films in Simulated Body Fluids submitted to Polymers Journal. This manuscript is generally well written and clearly presented however still needs to address some comments, and thus require substantial major revision to improve the quality of the manuscript.
- Abstract looks very general and not informative. In abstract authors should mention should mention the values of results and importance of research work in one or two sentences.
- In the introduction section, write the novelty of the work and the problem statement clearly. For the first para refer and cite Industrial Crops and Products 150, 112425, 2020; For line no 36 give details of strains
- Statistical analysis of the results should be provided in the materials and methods section. It's important for all experimental work Report these values in the results and discussion
- The reason of the gastric juice effectiveness need to discussed somewhere.
- In table 2 for PBS % of MW why higher in the first two weeks?
- Surprisingly there is no substantial discussion of results with the literature authors should concentrate on this issue during the revision stage.
- Techno Economic challenges and limitations of the developed system should be included?
- The conclusion of the study needs to be added with the specific output obtained from the study, it could be modified with precise outcomes with a take home message.
- Some English and grammar mistakes are present that need to be correct to improve the quality of the manuscript.
Author Response
Dear reviewer,
Thank you for Your time taken for reviewing the submitted manuscript entitled Degradation of P(3HB-co-4HB) films in Simulated Body Fluids, we appreciate the objectivity of Your comments and suggestions, and found the reviewed manuscript much improved.
We have properly considered each point of Your comments and hereby suggest accordingly modified the manuscript. Additionally, we would like to clarify (explain) some of Your points:
Point 2: In the introduction section, write the novelty of the work and the problem statement clearly. For the first para refer and cite Industrial Crops and Products 150, 112425, 2020; For line no 36 give details of strains
Response 2: The proposed reference has been applicated to “future prospects” discussion within the conclusion, which we found more suitable to cite. Thank You for the interesting reference inspiration. All copolymers were produced by Aneurinibacillus sp. H1 previously described in: (Pernicova et al. 2020) (available at https://doi.org/10.3390/polym12061235).
Point 4: The reason of the gastric juice effectiveness need to discussed somewhere.
Response 4: The effectiveness of SGJ lies in low pH of the medium. In correlation to individual analyses, t is discussed in sections 3.1, 3.2 and 3.3.
Resting points have been implemented directly to the manuscript. Also, the formal site and English has been checked and corrected by native speaker. Please, kindly find the modified manuscript enclosed.
We hope that provided modifications and offered explanations will meet Your requirements.
Best regards,
The authors collective

Reviewer 3 Report
- The research paper Degradation of P(3HB-co-4HB) films in Simulated Body Fluids is based on the research work reported on the synthesis of P3HB and copolymers. Although it has references to show the synthetic pathways and synthesis conditions, it is recommended to mention and show the synthetic pathways in the paper
- Incorporating 36 % and 66 % of 4HB resulted in 2 polymers, any other ratios were used to make polymers?
- For biological related studies majority of the polymers synthesized or in the process aim to degrade under neutral conditions, around pH 7. Here the polymer is susceptible to degradation under acidic conditions only.
- No degradation was observed in the first couple of weeks and very less/3-5% degradation was observed after 4 weeks. Major degradation was observed after only the 8th week. In terms of degradation rate, this is very slow. Need to study more to improve the degradation time.
- Seems only 1st DSC cycle was considered. Try in 3 cycles and compare the results. Show all the cycle results
- Polymer purity and reproducibility need to be presented
- reference format is inconsistent, especially the titles format
Author Response
Dear reviewer,
Thank you for Your time taken for reviewing the submitted manuscript entitled Degradation of P(3HB-co-4HB) films in Simulated Body Fluids, we appreciate the objectivity of Your comments and suggestions, and found the reviewed manuscript much improved.
We have properly considered each point of Your comments and hereby suggest accordingly modified the manuscript. Additionally, we would like to clarify (explain) some of Your points:
Point 1: The research paper Degradation of P(3HB-co-4HB) films in Simulated Body Fluids is based on the research work reported on the synthesis of P3HB and copolymers. Although it has references to show the synthetic pathways and synthesis conditions, it is recommended to mention and show the synthetic pathways in the paper
Response 1: The biosynthetic preparation of the polymer materials followed the previously optimized protocol – see ref. (Pernicova et al. 2020; Sedlacek et al. 2020) available at https://doi.org/10.3390/polym12061235 and https://doi.org/10.3390/polym12061298. The biosynthetic conditions and procedures are listed in “Materials and Methods”, section 2.1. Preparation of polymer films.
Point 2: Incorporating 36 % and 66 % of 4HB resulted in 2 polymers, any other ratios were used to make polymers?
Response 2: Stated copolymers were the sole copolymers used for this study. We consider these two ratios as sufficiently illustrative. The copolymers for degradation test were chosen based on our previous publications (Pernicova et al. 2020; Sedlacek et al. 2020) (available at https://doi.org/10.3390/polym12061235 and https://doi.org/10.3390/polym12061298), the choice of appropriate monomer ratios were based on the yields of biosynthetic process presented in the referred publication. Further, additional copolymers would complicate logistics of provided biodegradation tests and considerably prolong the research, while not offer adequate informative value.
Point 3: For biological related studies majority of the polymers synthesized or in the process aim to degrade under neutral conditions, around pH 7. Here the polymer is susceptible to degradation under acidic conditions only.
Response 3: This study is aimed to explore degradability of various PHA in simulated physiological conditions. For this purposes, we have tested biodegradability of prepared films in gastral system, in simulated gastric juice (SGJ, pH 1.6), artificial colonic fluid (ACF, pH 7.8) and phosphate buffer solution (PBS, pH 7.4) as a reference. The major purpose of the study was to illustrate variable degradation rates in the simulated gastral parts, show the affectability of the degradation rate (or mechanism) by simple handling of the chain composition (biosynthesis process) and to suggest tested materials as suitable for in vivo biomedical applications in gastral therapy. According to molecular weight changes and scanning electron micrograms, we have observed also apparent degradation of tested specimens in ACF. Actually, the degradation reached highest rapidity in SGJ, probably resourcing from the low pH and possible acidic catalysis. On the other hand, samples in ACF showed surface erosion typical for enzymatic degradation, therefore, we suggested different major mechanisms of degradation. Based on he obtained results, we propose tested copolymer films for analogic applications as reported e.g. here (Chang et al. 2014) (available at https://doi.org/10.1016/j.apsusc.2014.06.082) or here (Chang et al. 2012) (available at https://doi.org/10.1016/j.apsusc.2012.02.087) where PCL and PLA films were tested, respectively. The advantage of use of our model lays in the affectability of the biodegradation behaviour by the monomer composition handling.
Point 4: No degradation was observed in the first couple of weeks and very less/3-5% degradation was observed after 4 weeks. Major degradation was observed after only the 8th week. In terms of degradation rate, this is very slow. Need to study more to improve the degradation time.
Response 4: As stated below, the main purpose of the study was to prove the affectability of the biodegradation rate by the monomer composition. This hypothesis has been successfully approved. In addition, the degradation courses of PHA are generally slower in simulated physiological conditions than in environments ideal for the degradation. Human physiologic environment does not contain specific depolymerases for PHA but contains less specific esterases and lipases capable of hydrolysation of ester bond between monomers. Hence, the rate of biodegradation might be lower.
Point 5: Seems only 1st DSC cycle was considered. Try in 3 cycles and compare the results. Show all the cycle results
Response 5: We have provided only first DSC cycle scans (all scans are displayed). As far as the different rates of biodegradation were explained also by considering different initial crystallinity of the films, the DSC assay was designed to illustrate the initial crystallinity of the solvent-casted films after incubation in SBFs. The subsequent freezing-thawing cycles would better illustrate the inherent tendency of material to crystallize under defined conditions, nevertheless, it would not be relevant for the evaluation of the biodegradation rates.
Point 6: Polymer purity and reproducibility need to be presented
Response 6: The polymer purity was checked by Fourier-transform infrared spectroscopy (FTIR) as reported in previous work (Pernicova et al. 2020; Sedlacek et al. 2020) as referred above. There were no impurities detected, which corresponds to the stated previous works. Since this consistency and fully identical methodology, we consider the biosynthetic step as reproducible with stable purity of the final product. The biosynthetic methodology is also referred to the stated previous works within the manuscript.
Resting points have been implemented directly to the manuscript. Also, the formal site and English has been checked and corrected by native speaker. Please, kindly find the modified manuscript enclosed.
We hope that provided modifications and offered explanations will meet Your requirements.
Best regards,
The authors collective

Round 2
Reviewer 2 Report
My recommendation is
The authors have substantially revised the manuscript according to the comments.
The present form of the manuscript can be accepted for publication.
Author Response
Dear reviewer,
Thank You for Your specific comments that have contributed to the manuscript improvement. Please, kindly find the last version of the manuscript enclosed.
Best wishes,
The collective of authors
